# Type of PaperY192 within the SH2 Domain of Lck Regulates TCR Signaling Downstream of PLC-γ1 and Thymic Selection

**DOI:** 10.3390/ijms23137271

**Published:** 2022-06-30

**Authors:** Matthias Kästle, Camilla Merten, Roland Hartig, Carlos Plaza-Sirvent, Ingo Schmitz, Ursula Bommhardt, Burkhart Schraven, Luca Simeoni

**Affiliations:** 1Institute of Molecular and Clinical Immunology, Medical Faculty, Otto-von-Guericke University, 39120 Magdeburg, Germany; matthias_kaestle@gmx.de (M.K.); camilla.merten@med.ovgu.de (C.M.); roland.hartig@med.ovgu.de (R.H.); carlos.plazasirvent@rub.de (C.P.-S.); ingo.schmitz@rub.de (I.S.); ursula.bommhardt@med.ovgu.de (U.B.); 2Department of Molecular Immunology, Ruhr-University Bochum, 44801 Bochum, Germany; 3Health Campus Immunology, Infectiology and Inflammation (GC-I3), Medical Faculty, Otto-von-Guericke University, 39120 Magdeburg, Germany; 4Center for Health and Medical Prevention (CHaMP), Otto-von-Guericke University, 39120 Magdeburg, Germany

**Keywords:** Lck, T-cell development, thymic selection, TCR signaling, LckY192

## Abstract

Signaling via the TCR, which is initiated by the Src-family tyrosine kinase Lck, is crucial for the determination of cell fates in the thymus. Because of its pivotal role, ablation of Lck results in a profound block of T-cell development. Here, we show that, in addition to its well-known function in the initiation of TCR signaling, Lck also acts at a more downstream level. This novel function of Lck is determined by the tyrosine residue (Y192) located in its SH2 domain. Thymocytes from knock-in mice expressing a phosphomimetic Y192E mutant of Lck initiate TCR signaling upon CD3 cross-linking up to the level of PLC-γ1 phosphorylation. However, the activation of downstream pathways including Ca^2+^ influx and phosphorylation of Erk1/2 are impaired. Accordingly, positive and negative selections are blocked in Lck^Y192E^ knock-in mice. Collectively, our data indicate that Lck has a novel function downstream of PLCγ-1 in the regulation of thymocyte differentiation and selection.

## 1. Introduction

Developing T cells in the thymus progress through sequential intermediate stages, which are defined by the expression of the coreceptors CD4 and CD8 [1]. Most immature thymocytes lack the expression of both CD4 and CD8 (double-negative or DN thymocytes). During their maturation, DN thymocytes upregulate CD4 and CD8 and become double-positive (DP) thymocytes, which in turn undergo tightly regulated selection processes and finally become single positive (SP) thymocytes expressing either CD4 or CD8. A major drive for the progression through the different developmental stages is executed by signals that emanate from the preTCR and the mature αβTCR [2]. The preTCR includes a fully rearranged β-chain that pairs with an invariant preTα chain and is expressed on a subset of DN thymocytes known as DN3. Signaling from the preTCR drives the differentiation (β selection) of DN3 thymocytes to DN4 and subsequently to the DP stage [2,3]. At the DP stage, thymocytes express fully rearranged TCRα and TCRβ chains, resulting in the expression of a mature TCR. The TCR dictates the final maturation into SP thymocytes [1,2,4]. If the TCR fails to recognize self-peptide–MHC complexes, DP thymocytes die via death by neglect. Similarly, DP thymocytes binding self-peptide–MHC molecules with high affinity die by negative selection. Only DP cells expressing TCRs with low/intermediate affinity for self-peptide–MHC complexes are able to mature into SP thymocytes.

Several studies have established that both preTCR and TCR signaling in thymocytes are initiated by Lck, a member of the Src family of protein tyrosine kinases [5,6], and by the subsequent activation of downstream molecules of the canonical TCR signaling pathway [2]. A number of studies using different mouse models have shown the importance of Lck in T-cell development [7,8,9,10,11]. Lck^−/−^ mice display a marked thymic atrophy with only 10% of normal cellularity and a developmental block at the DN3 stage [7]. Both DP and SP thymocytes are barely detectable in the thymi of Lck^−/−^ mice, and accordingly, only very few T cells are found in the periphery. The defect in the generation of mature T cells appears to be due to an impaired TCR signaling, as suggested by the defective phosphorylation of TCRζ, Zap70, PLC-γ1, and Vav [12,13].

Additional overexpression studies further corroborated the importance of Lck in thymic development. The transgenic expression of Lck under the control of the Lck proximal promoter element, which is transcriptionally active at the DN stage [14], disrupts thymic development, resulting in significantly reduced numbers of DP thymocytes and mature T cells [15]. On the other hand, the transgenic expression of a dominant negative catalytically inactive Lck K273R mutant also expressed under the control of the proximal Lck promoter arrested thymic development at the DN3 stage [16]. The role of Lck at the DP stage of thymic development was assessed using transgenic mice in which Lck expression is controlled by the Lck distal promoter element [17], which is active at the DP stage [14]. In these mice, the expression of a constitutively active Y505F Lck mutant at the DP stage increased the generation of SP thymocytes [17]. In addition, the Y505F transgene was also able to drive SP maturation in the absence of TCR engagement. Collectively, these data have demonstrated the critical role of Lck at all stages of T-cell development.

More recent data have shown that mice expressing a Y192E Lck mutant display a defective thymic development [18,19]. Nevertheless, it is still unclear how this particular mutant of Lck affects thymic development. Y192 lies within the SH2 domain of Lck, and initial studies suggest that phosphorylation of Y192 alters the affinity of the SH2 domain for Lck binding partners [20,21]. More recently, it has been proposed that phosphorylation of Y192 affects the interaction between Lck and CD45 [18]. CD45 positively regulates Lck activity by dephosphorylating Y505, which is located in the negative regulatory tail of Lck [22,23,24]. Hence, it was proposed that loss of Lck/CD45 interactions upon phosphorylation of Y192 results in the hyperphosphorylation of Y505 and, consequently, in the inactivation of Lck [18]. This idea was supported by the observation that both Jurkat T cells and murine primary T cells expressing Lck^Y192E^ display a block in TCR signaling and impaired proliferation [18]. However, most recent data from our laboratory have shown that despite the hyperphosphorylation on Y505 and the inability to interact with CD45, Lck^Y192E^ retains its enzymatic activity in vitro. Perhaps more importantly, even in cells lacking CD45, Lck^Y192E^ was unable to fully activate Zap70 [19]. Therefore, we assumed that the defective thymic development observed in retrogenic mice expressing Lck^Y192E^ [18] and in Lck^Y192E^ knock-in mice [19] is not exclusively due to the loss of Lck activity.

In line with this hypothesis, we here show that Lck^Y192E^ is still capable of initiating membrane proximal TCR signaling in thymocytes until the level of PLC-γ1. However, the TCR is uncoupled from the activation of further downstream signaling events, such as the induction of Ca^2+^ influx or the activation of Erk1/2. Accordingly, both positive and negative selections are reduced in Lck^Y192E^ mice. Taken together, our data suggest that Lck regulates the activation of canonical prosurvival/differentiation pathways downstream of PLC-γ1, which are required for the selection of developing T cells via Y192.

## 2. Results

### 2.1. Defective Thymocyte Selection in Lck^Y192E^ Mice

We have recently shown that TCR stimulation induces phosphorylation of Y192 in murine splenic T cells [19]. Therefore, we first assessed whether Y192 is also phosphorylated in thymocytes. To this end, isolated murine thymocytes were either left unstimulated or were stimulated with CD3 antibodies. Subsequently, the phosphorylation of Y192 was analyzed using a phospho-specific antibody, and the phosphorylation of Erk1/2 was used to monitor efficient activation of the TCR signaling cascade (Figure 1A). Figure 1A,B shows that Y192 is constitutively phosphorylated in thymocytes and that TCR stimulation results in a sustained augmentation in the phosphorylation of Y192, similar to what we observed in splenic T cells [19]. These data indicate that phosphorylation of Y192 may play an important role in TCR signaling and, hence, thymocyte development. Indeed, we have previously shown that knock-in mice expressing a phosphomimetic Lck^Y192E^ display an altered T-cell development [19].

In order to shed light on how Lck^Y192E^ affects thymocyte development, we performed a detailed phenotypical characterization of the thymus of Lck^Y192E^ knock-in mice. In line with our previous observation [19], Figure 2 shows that Lck^Y192E^ mice display a strong decrease in total thymocyte numbers (Figure 2A) and an alteration in the distribution of thymocyte subsets (Figure 2B). Despite the fact that the proportion of DN cells was significantly increased (Figure 2B), the total number of DN cells was largely comparable between Lck^Y192E^ and wild-type (WT) mice (Figure 2C). We further analyzed DN subsets from Lck^Y192E^ mice and WT controls (Figure 3A). Lck^Y192E^ mice displayed elevated numbers of DN3 and reduced numbers of DN4 cells (Figure 3B), consistent with a developmental arrest at the DN3 stage resulting from defective preTCR signaling.

The Y192E mutation also significantly affected later developmental stages as the numbers of DP and SP thymocytes were strongly decreased in Lck^Y192E^ mice compared with WT controls (Figure 2B,C). Therefore, we further evaluated whether the selection processes of DP cells are affected by the Y192E mutation. Selection processes can be indirectly monitored by analyzing the expression of cell surface markers, such as CD69 and TCRβ [25,26,27,28]. To this end, cells were stained with CD69 and TCRβ antibodies to identify pre- and postselection thymocytes according to the scheme depicted in Figure 4A [29]. We found that thymocytes from Lck^Y192E^ mice accumulate at a transitional stage characterized by intermediate levels of TCRβ and low or no CD69 expression (Figure 4B,C). Accordingly, the proportions of postselected TCR^hi^ CD69^hi^ thymocytes were markedly reduced in the knock-in mice (Figure 4B,C). In addition to the upregulation of TCRβ and CD69, thymocytes undergoing selection also upregulate CD5 [30]. In agreement with the data shown above, indicating defective upregulation of CD69 and TCRβ, we also noted that the upregulation of CD5 was reduced in thymocytes of Lck^Y192E^ knock-in mice (Figure 4D). These observations corroborate the hypothesis that thymic selection is significantly altered in Lck^Y192E^ mice.

To better understand how the phosphomimetic Lck^Y192E^ mutation affects thymocyte differentiation and selection, we crossed Lck^Y192E^ mice with mice carrying transgenic TCRs. TCR transgenic mice allow the efficient monitoring of the outcome of thymic selection in a synchronized monoclonal population carrying the same TCR. We first assessed positive selection using the MHC-II-restricted OT-II transgenic TCR (Vα2Vβ5.1). Figure 5A shows that Lck^Y192E^ knock-in mice expressing the OT-II TCR display a markedly reduced thymic cellularity and a strong increase in DN cells concomitant with a reduced proportion of DP cells. This observation is in line with a block at the DN3 to DN4 transition that we also observed in non-TCR transgenic mice. In addition to the reduced thymic cellularity, we observed that OT-II Lck^Y192E^ knock-in mice display a strongly reduced proportion of CD4^+^ thymocytes (Figure 5B). To specifically monitor the development of OT-II transgenic thymocytes, we performed flow cytometry analyses using an antibody directed against the transgenic TCRVβ5.1 chain. We found that OT-II TCR transgenic thymocytes expressing Lck^Y192E^ failed to upregulate the transgenic TCRβ chain (Figure 5C), thus suggesting a strong defect in positive selection. Accordingly, positively selected TCRVβ5.1^+^ CD4^+^ SP thymocytes were barely detectable in Lck^Y192E^ knock-in mice (Figure 5D).

To study the positive selection of MHC-I-restricted T cells and to corroborate the data shown above, we crossed Lck^Y192E^ mice with mice expressing the HY male-specific antigen TCR. Because of the lack of expression of the HY male antigen, HY transgenic thymocytes are positively selected toward the CD8^+^ lineage in female mice [31]. Again, thymic cellularity was strongly reduced in female HY Lck^Y192E^ knock-in mice (Figure 6A). Analyses using the clonotypic antibody T3.70 specific for the HY TCR revealed comparable expression levels between WT and knock-in mice (Figure 6B). However, the positive selection of T3.70^+^CD8^+^ SP thymocytes was blocked in female HY Lck^Y192E^ mice (Figure 6C). Collectively, the data shown above indicate that positive selection is strongly impaired in thymocytes expressing the Y192E mutation.

Taking advantage of the HY mouse model, we also investigated the function of the Y192E Lck mutation in negative selection. Transgenic thymocytes expressing the HY male-specific antigen TCR are deleted (i.e., negatively selected) in male mice, thus resulting in a hypocellular thymus. The expression of the Y192E mutation had no effect on thymic cellularity in male animals (Figure 6D) and also did not significantly change the expression of the transgenic TCR (Figure 6E). Nevertheless, the effect of the Y192E mutation on negative selection was striking. As shown in Figure 6F, we indeed detected a high proportion of DP thymocytes in male HY Lck^Y192E^ mice, which we did not observe in male control mice, as DP thymocytes are deleted in these animals. These data indicate that, in addition to positive selection, also negative selection is impaired in Lck^Y192E^ mice.

### 2.2. Lck^Y192E^ Initiates TCR Signaling in Thymocytes

One reason for the impaired differentiation and selection of thymocytes could be that Lck^Y192E^ is enzymatically less active than Lck^WT^ and, hence, not able to signal properly. This hypothesis is in line with the observation that Lck^Y192E^ mice display a hypocellular thymus similar to Lck^−/−^ mice [7,9]. However, we have most recently shown by in vitro kinase assays that Lck^Y192E^ displays normal enzymatic activity in thymocytes [19], despite the hyperphosphorylation on Y505, which was 23 to 490 times stronger than in wild-type mice (Figure 7A). Thus, the defects observed in the Lck^Y192E^ mice appear not to be due to a decreased enzymatic activity of the Lck mutant. Therefore, we next assessed whether and to which extent TCR signaling is initiated in thymocytes expressing Lck^Y192E^. We found that global tyrosine phosphorylation and the phosphorylation of Zap70 (Y319), LAT (Y191), and PLC-γ1 (Y783) were readily inducible upon TCR stimulation in Lck^Y192E^ thymocytes (Figure 7B). This strongly suggests that Lck^Y192E^ is capable of initiating TCR signaling. The data represented in Figure 7B show that signaling is even stronger in Lck^Y192E^ thymocytes compared with Lck^WT^ thymocytes. However, because of the increased expression of the TCR/CD3 complex on thymocytes of Lck^Y192E^ mice (Figure 7C), as well as the fact that Lck^Y192E^ expression is higher than Lck^WT^ (Figure 7A), we are hesitant to propose that TCR signaling is enhanced in Lck^Y192E^ mice.

### 2.3. The TCR Is Uncoupled from the Activation of Pathways Downstream of PLC-γ1 in Lck^Y192E^ Thymocytes

Despite its ability to initiate TCR signaling, Lck^Y192E^ does not support thymocyte differentiation. This observation suggests that the proximal TCR signaling is not properly transmitted to the activation of further downstream pathways. To explore this hypothesis, we investigated the activation of signaling pathways downstream of PLC-γ1, such as Ca^2+^ influx and phosphorylation of Erk1/2, which are known to regulate thymic selection processes [32,33,34,35,36,37,38]. We first analyzed Ca^2+^ influx upon TCR stimulation by flow cytometry, and we observed that thymocytes from Lck^Y192E^ mice showed a reduced TCR-induced Ca^2+^ influx (Figure 7D). We next assessed whether the activation of Erk1/2 is affected in Lck mutant mice. Figure 7E shows that, despite the augmented expression of the TCR/CD3 complex and the intact proximal signaling, the TCR-induced phosphorylation of Erk1/2 is reduced in thymocytes from Lck^Y192E^ mice. We also assessed Erk1/2 activation by intracellular staining using a phospho-specific Erk1/2 antibody in ex vivo isolated thymocytes at different stages of thymocyte selection. Figure 7F shows that the transitional population of thymocytes, which strongly accumulates in Lck^Y192E^ mice (Figure 4B,C), contains a reduced proportion of phospho-Erk1/2 expressing cells (Figure 7F). These data indicate that pathways distal to the triggered TCR and distal to PLC-γ1 are downregulated in thymocytes from Lck^Y192E^ mice.

Collectively, these data suggest that Lck^Y192E^ efficiently initiates TCR signaling in developing thymocytes but is uncoupled to the activation of further downstream signaling events required for thymocyte differentiation.

## 3. Discussion

Many studies have shown that loss of Lck has a major impact on T-cell development, because thymocytes from Lck^−/−^ mice cannot activate differentiation pathways due to impaired preTCR and TCR signaling [7,9,13]. Previous observations from Courtney and colleagues [18] and from our group [19] indicated that mice expressing an Lck^Y192E^ mutant display an impaired DN3 to DN4 transition, thus corroborating the important role of Lck in early stages of thymocyte development. In this study, we additionally found that Lck^Y192E^ mice show a block of positive and negative selection. Because of the defect at different stages of thymic development in Lck^Y192E^ mice, it appears logical to assume that Lck^Y192E^ has lost its enzymatic activity and the ability to initiate preTCR and TCR signaling. This hypothesis is supported by the observation that Lck^Y192E^ shows a reduced association with CD45 and is hyperphosphorylated on the negative regulatory Y505 [18]. Nevertheless, we surprisingly found that Lck^Y192E^ still retains its enzymatic activity in thymocytes [19], despite being hyperphosphorylated on Y505 and having lost the interaction with CD45. Moreover, Lck^Y192E^ can initiate TCR signaling in thymocytes. Similar observations were reported for CD45^−/−^ mice. Indeed, in CD45^−/−^ thymocytes, Lck is hyperphosphorylated on Y505, retains its enzymatic activity, and can initiate TCR signaling [39,40]. Thus, Lck activity in thymocytes is not exclusively dependent on CD45 and Y505 dephosphorylation.

Our biochemical characterization of signaling events in Lck^Y192^ thymocytes showed that TCR signaling proceeds up to the phosphorylation of PLC-γ1 on Y783. However, pathways downstream of PLC-γ1, such as Ca^2+^ influx and the activation of Ras-Erk1/2, are impaired. Interestingly, similar results were observed in knock-in mice expressing Lck^W97A^ [41], a mutation that has previously been shown to disrupt the function of the SH3 domain of Lck [42]. Similar to Lck^Y192E^ mice, Lck^W97A^ knock-in mice display defective positive and negative selection [41]. In addition, TCR signaling is grossly unaffected until the phosphorylation of PLC-γ1. However, the activation of Erk1/2 is strongly decreased. Thus, from our study and from the study of Rudd and coworkers, it clearly emerges that Lck plays a role not only in the initiation of TCR signaling but also in the regulation of downstream signaling pathways and that both the SH2 and SH3 domains of Lck are involved. How Lck performs this function remains elusive. In an attempt to shed light on this issue, we performed mass spectrometry analyses of Lck immunoprecipitated from cells expressing either Lck^WT^ or Lck^Y192E^ [19]. Despite this effort, we did not observe differences between immunoprecipitates from Lck^WT^ and Lck^Y192E^ expressing cells.

Lck could function at different levels downstream of PLC-γ1. First, Lck could regulate the Ras cascade by contributing to the activation of Raf-1 [43]. Second, Lck can also regulate Ca^2+^ influx by interacting with the type I IP_3_ receptor. Indeed, loss of Lck expression or activity resulted in IP_3_ receptor downregulation [44]. Finally, recent data suggest that Lck interacts with SUMO-specific protease 1 (SENP1) and that SENP1 is an Lck substrate [45]. Sumoylation plays an important role in the regulation of TCR signaling [46,47,48,49]. Therefore, it is possible that Lck by controlling the activity of SENP1 may also regulate TCR signaling at multiple levels by modulating the sumoylation system.

In conclusion, in this study we revealed a new function of Lck, which appears to regulate thymocyte maturation by regulating pathways downstream of PLC-γ1. This function of Lck is determined by the tyrosine residue Y192 located in its SH2 domain.

## 4. Materials and Methods

### 4.1. Antibodies

The following antibodies were purchased from Cell Signaling Technologies (Danvers, Massachusetts, USA): pLat (Y191), pPLC-γ1 (Y783), pp44/42 MAPK (pErk1/2) (T202/Y204), pZap70 (Y319), pLck (Y505), pSrc (Y416), and Erk1/2. The antibody against β-actin was purchased from Sigma-Aldrich, whereas the antibody against Lck (clone 3A5) was purchased from Santa Cruz Biotechnology. The antibodies for the flow cytometry studies were obtained from BD Pharmingen/Biosciences (Franklin Lakes, NJ, USA) and BioLegend (San Diego, CA, USA). pY (4G10) was produced by a hybridoma.

### 4.2. Mice

Mice were kept in a specific pathogen-free facility at the Medical Campus of the University Magdeburg according to the German animal law. Lck^Y192E^ knock-in mice were generated by Dr. M. Herold and Dr. A. Kueh (WEHI, Melbourne, Australia) using CRISPR/Cas technology. The obtained heterozygous mice were backcrossed to a C57BL/6J genetic background. Lck^Y192E^ mice were interbred with OT-II [50] and HY [51] TCR transgenic mice.

### 4.3. Stimulation and Lysis of Cells

2.5 × 10^6^ to 5 × 10^6^ murine thymocytes were stimulated with 5–10 µg/mL biotinylated CD3ε 145–2C11 antibody (BioLegend, San Diego, CA, USA), followed by cross-linking with 20 µg/mL neutravidin at 37 °C. Alternatively, cells were stimulated with biotinylated CD3ε antibody immobilized on superavidin-coated microbeads (Bang Laboratories), as previously described [52]. Cells were lysed in 1% LM (lauryl maltoside), 1% NP-40, 1 mM phenylmethylsulfonyl fluoride, 10 mM NaF, 10 mM EDTA, 50 mM Tris-HCl (pH 7.5), and 150 mM NaCl.

### 4.4. Immunoblotting

Samples were assayed using SDS-PAGE. Proteins were transferred (semidry) on a polyvinylidene difluoride membrane (Amersham/GE Healthcare, Chicago, IL, USA). Membranes were blocked in 5% milk and incubated with primary antibodies in 5% milk or 5% BSA for 1 h. Secondary antibodies coupled with a fluorophore (LI-COR, Lincoln, NE, USA)) or horseradish peroxidase (Dianova/Jackson ImmunoResearch Laboratories, Inc., West Grove, PA, USA) were diluted in 5% milk. To detect the protein signals on the membranes, an Odyssey infrared imager (LI-COR, Lincoln, NE, USA) or ECL (Amersham/GE Healthcare; Chicago, IL, USA was used. The densiometric analyses of the blots were performed with the image software Image Studio (LI-COR, Lincoln, NE, USA)). The total median values of densiometric analysis were used for quantifications.

### 4.5. Flow Cytometry Measurements

Thymocytes were isolated using a 70 µm cell strainer (Falcon Corning Lifescience, Tewksbury, MA, USA). Antibodies (BD Biosciences, Franklin Lakes, NJ, USA or BioLegend, (San Diego, CA, USA) were diluted 1:100 in PBS for each staining and added to the cells (1 × 10^6^ cells/sample). Samples were measured with a BD Fortessa I (3 Lasers) and BD Calibur using the CellQuest and BD Diva software (BD Biosciences, Franklin Lakes, NJ, USA). The data were analyzed using FlowJo software (BD Biosciences, Franklin Lakes, NJ, USA).

### 4.6. Calcium Flux

Thymocytes were loaded with Indo-1 (Thermo Fisher Scientific, Waltham, MA, USA) for 45 min at 37 °C in RPMI 1640 without phenol red (Gibco Thermo Fisher Scientific, Waltham, MA, USA) and washed with RPMI without phenol red for 45 min at 37 °C. Stimulation was induced by the addition of neutravidin cross-linked CD3ε (145-2C11; BioLegend, San Diego, CA, USA). As a positive control, 100 nM ionomycin (Sigma-Aldrich, Saint Louis, MO, USA) was added 8–10 min after Ab stimulation. Ca^2+^ influx was measured with a BD Fortessa II (BD Biosciences, Franklin Lakes, NJ, USA) using a 325 nm laser line of a helium cadmium laser. Emission wavelength ranges from 395 and 500 to 520 nm were detected, and the ratio of the two emission intensities was calculated and analyzed with FlowJo (BD Biosciences, Franklin Lakes, NJ, USA).

### 4.7. Intracellular Staining

2 × 10^6^ thymocytes were left for 10 min at 37 °C and stained with CD4-PE-Cy5, CD8-APC-Cy7, CD5-FITC, CD69-PE, and TCRβ-Pacific Blue (BioLegend, San Diego, CA, USA). Subsequently, cells were fixed and permeabilized for 10 min with 3.7% PFA (ChemCruz Biochemicals, Santa Cruz, CA, USA) + 0.2% saponin (Sigma Aldrich, Saint Louis, MO, USA). Subsequently, samples were washed and incubated with a phospho-specific Erk1/2 antibody (Cell Signaling, Danvers, MA, USA) in PBS containing 0.1% saponin, 0.1% sodium azide, and 0.2% BSA. Cells were washed and stained with APC-conjugated goat anti-rabbit secondary antibody (Jackson ImmunoResearch Laboratories, Inc., West Grove, PA, USA). Samples were analyzed on a FACS Fortessa I (BD Biosciences, Franklin Lakes, NJ, USA).

### 4.8. Statistics

Statistical analyses were performed using GraphPad Prism software. Unless otherwise indicated, statistical significance was determined between groups using an unpaired Student′s *t* test. The minimum acceptable level of significance was *p* ≤ 0.05.

## Figures and Tables

**Figure 1 ijms-23-07271-f001:**
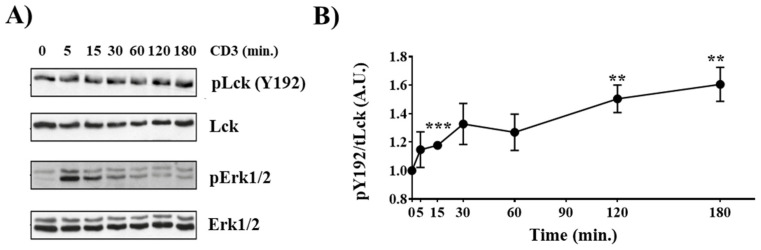
**Inducible Y192 phosphorylation upon TCR stimulation in thymocytes.** (**A**) Thymocytes were stimulated with immobilized CD3 antibodies for the indicated time points. Subsequently, cell lysates were analyzed for Y192 phosphorylation using a phospho-specific antibody. Successful stimulation was monitored using a phospho-Erk1/2 antibody. Immunoblotting using total Lck and Erk1/2 antibody was used to show equal loading. (**B**) The graph depicts densiometric analysis of the Y192 phospho-signal normalized to that of total Lck. Dots represent mean values ± SEM of three independent mice. Statistical analyses were conducted using an unpaired Student’s *t* test (** *p* ≤ 0.01, *** *p* ≤ 0.001; where not indicated, the values were not statistically significant).

**Figure 2 ijms-23-07271-f002:**
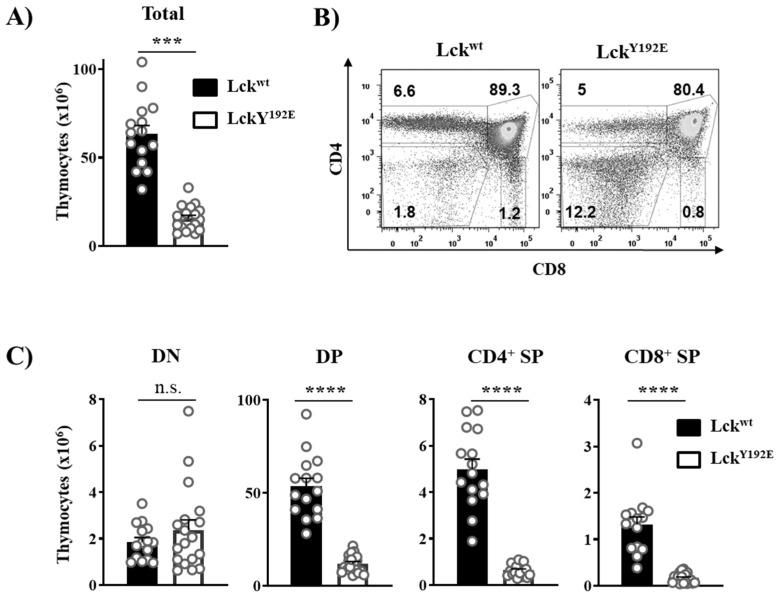
**Lck^Y192E^ mice display a severe defect in T-cell development.** (**A**) Total numbers of thymocytes isolated from Lck^wt^ and Lck^Y192E^ animals. (**B**) Isolated thymocytes were stained with CD4 and CD8 antibodies to distinguish different thymic subpopulations. Numbers indicate percentages of the cells in each gate. One representative dot plot is shown. (**C**) Bar graphs represent summary statistics of the cell number in each thymic subpopulation. Each dot shown in (**A**,**C**) represents one mouse, whereas columns show mean values of all analyzed mice + SEM (*n* = 15–18). Statistical analyses were conducted using an unpaired Student’s *t* test (*** *p* ≤ 0.001, **** *p* ≤ 0.0001, n.s. = not statistically significant).

**Figure 3 ijms-23-07271-f003:**
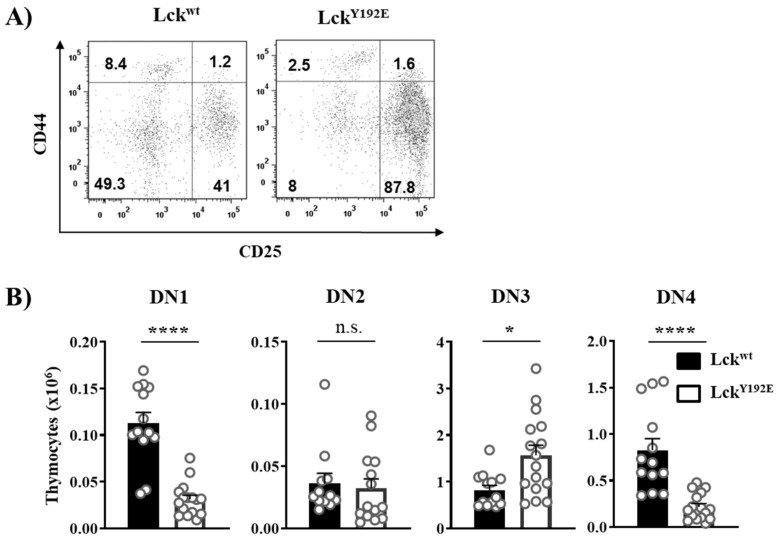
**Developmental block at the β selection checkpoint in Lck^Y192E^ mice.** (**A**) To identify the four stages of DN thymocytes, isolated cells were stained with CD4, CD8, CD44, and CD25 antibodies. Cells were gated on DN thymocytes, and CD44 and CD25 expressions were analyzed. Numbers indicate percentages of the cells in each quadrant. One representative dot plot is shown. (**B**) Bar graphs represent summary statistics of the cell number in each DN subset, which are defined as: DN1 (CD25^−^/CD44^+^), DN2 (CD25^+^/CD44^+^), DN3 (CD44^−^/CD25^+^), and DN4 (CD44^−^/CD25^−^). Each dot shown represents one mouse, whereas columns show mean values of all analyzed mice + SEM (*n* = 12–15). Statistical analyses were performed using an unpaired Student’s *t* test (* *p* ≤ 0.05, **** *p* ≤ 0.0001, n.s. = not statistically significant).

**Figure 4 ijms-23-07271-f004:**
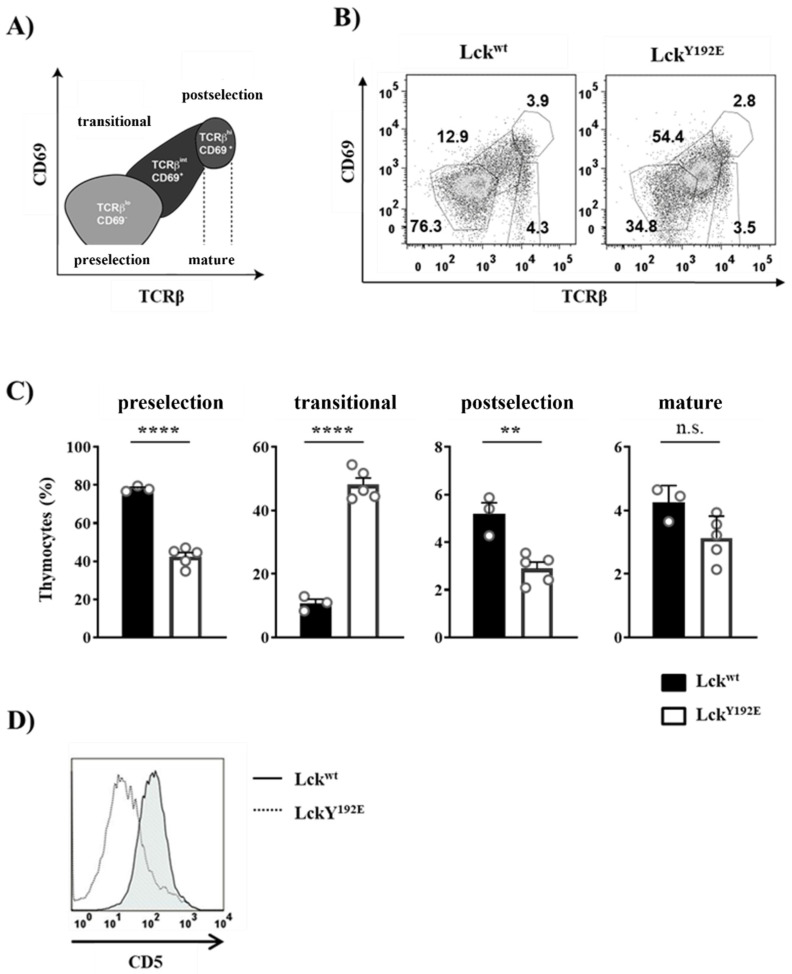
**Altered transition of Lck^Y192E^ thymocytes through the developmental stages.** (**A**) Schematic representation of developmental stages during thymocyte maturation defined according to TCRβ and CD69 expression. Preselection thymocytes were defined as cells expressing low levels of TCRβ and no CD69 (TCRβ^l^°CD69^−^), whereas postselection thymocytes were characterized by high levels of TCRβ and the expression of CD69 (TCRβ^hi^CD69^+^). Thymocytes expressing intermediate levels of TCRβ and CD69 represent a transitional population (TCRβ^int^CD69^+^). Mature thymocytes express high levels of TCRβ and low or no CD69 expression. (**B**) Thymocytes were isolated from either Lck^wt^ or Lck^Y192E^ mice and stained with antibodies against TCRβ and CD69. Numbers indicate percentages of the cells in each gate. One representative dot plot is shown. (**C**) Bar graphs represent summary statistics of the cell distribution in each developmental stage. Each dot represents one mouse, whereas columns show mean values of all analyzed mice + SEM (*n* = 3–5). Statistical analyses were conducted using an unpaired Student’s *t* test (** *p* ≤ 0.01, **** *p* ≤ 0.0001, *n*.s. = not statistically significant). (**D**) Thymocytes were stained with CD4, CD8, and CD5 antibodies. One representative histogram overlay depicting CD5 expression on DP-gated cells from either Lck^wt^ or Lck^Y192E^ mice (*n* = 7) is shown.

**Figure 5 ijms-23-07271-f005:**
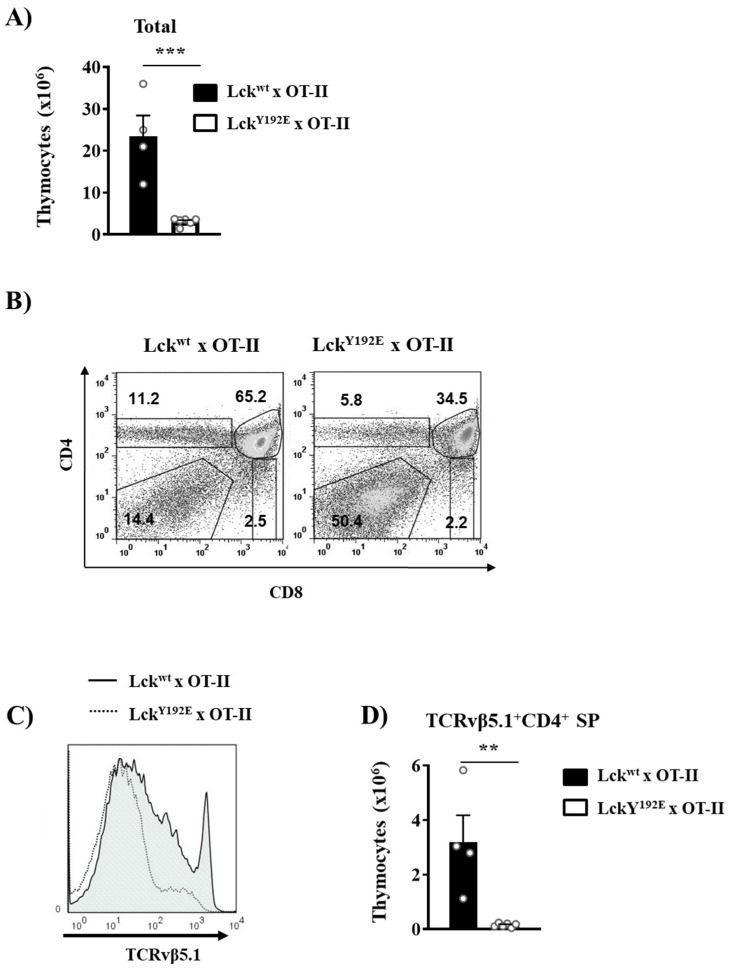
**Defective positive selection in OT-II TCR transgenic mice expressing Lck^Y192E^.** (**A**) Total numbers of thymocytes isolated from Lck^wt^ and Lck^Y192E^ x OT-II animals. (**B**) Thymocytes were isolated from the indicated mouse strains and stained with CD4 and CD8 antibodies. Numbers indicate percentages of the cells in each gate. One representative dot plot is shown (*n* = 4–6). (**C**) Representative histogram overlay showing the expression of the transgenic TCRVβ5.1 on thymocytes from either Lck^wt^ or Lck^Y192E^ mice (*n* = 4–6). (**D**) Bar graphs represent summary statistics of the numbers of positively selected CD4^+^ SP thymocytes expressing the transgenic TCRVβ5.1 chain. Each dot represents one mouse, whereas columns show mean values of all analyzed mice + SEM (*n* = 4–6). Statistical analyses were performed using an unpaired Student’s *t* test (** *p* ≤ 0.005, *** *p* ≤ 0.001).

**Figure 6 ijms-23-07271-f006:**
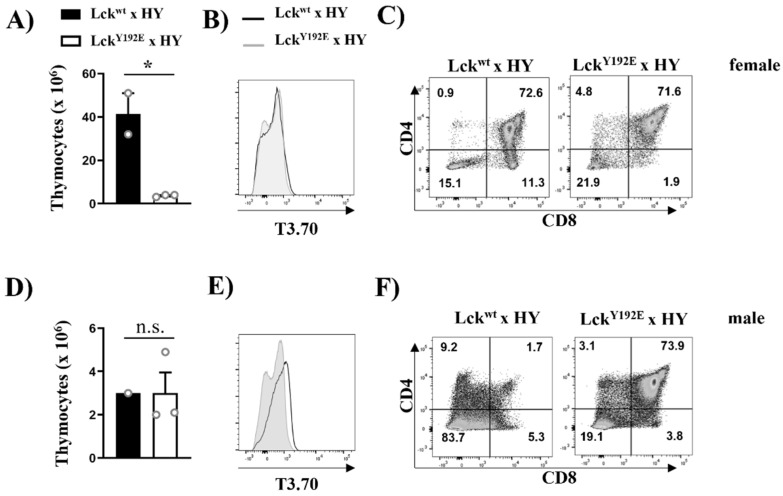
**Defective positive and negative selection in HY TCR transgenic mice expressing Lck^Y192E.^.** Thymocytes were isolated from female (**A**–**C**) or male (**D**–**F**) HY TCR transgenic mice expressing either Lckwt or LckY192E. Bar graphs in (**A**,**D**) represent summary statistics of total thymocyte numbers. Each dot represents one mouse, whereas columns show mean values of all analyzed mice + SEM (*n* = 1–3). Statistical analyses were performed using an unpaired Student’s *t* test (* *p* ≤ 0.05, n.s. = not statistically significant). (**B**,**E**) Representative histogram overlay showing the expression of the transgenic HY-TCR (T3.70) on thymocytes from either Lck^wt^ (filled histograms) or Lck^Y192E^ (empty histograms) mice (*n* = 1–3). (**C**,**F**) Dot plots showing CD4 and CD8 expression on T3.70^+^-gated thymocytes. Numbers indicate percentages of the cells in each quadrant. One representative dot plot is shown (*n* = 1–3).

**Figure 7 ijms-23-07271-f007:**
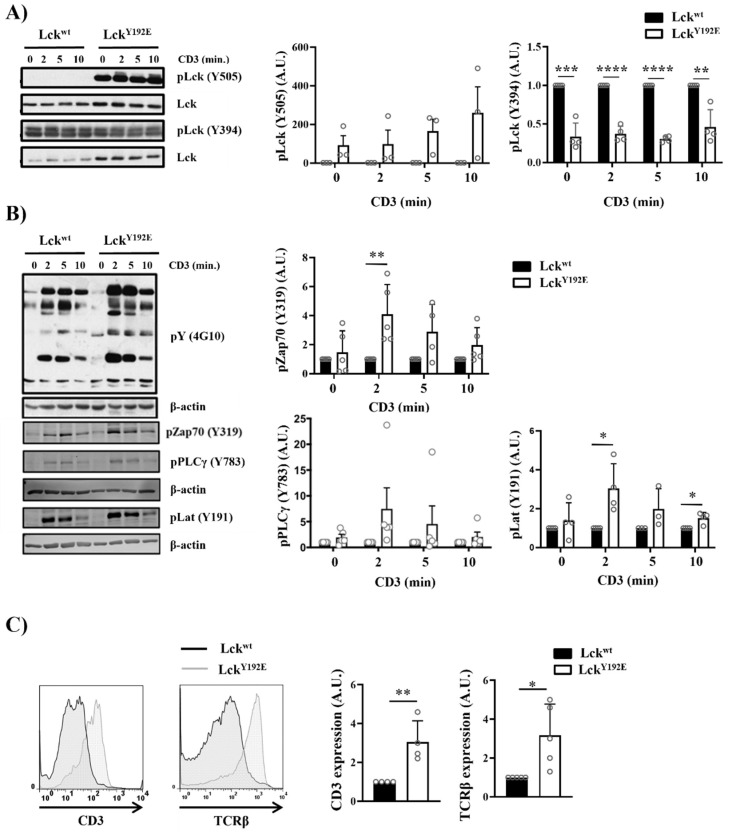
**Intact proximal TCR signaling but impaired activation of downstream pathways in stimulated thymocytes from Lck^Y192E^ knock-in mice.** (**A**,**B**) Thymocytes from Lck^wt^ and Lck^Y192E^ mice were stimulated with CD3 antibodies for the indicated time points. Subsequently, cell lysates were prepared, and immunoblot analyses were performed using the indicated antibodies. β-actin was used to show equal loading. One representative immunoblot is shown on the left side of the figure (*n* = 3–5). Graphs depicted on the right side in (**A**,**B**) show statistical analyses of the immunoblots. Bands of the immunoblots were quantified using the Image Studio software, and values were normalized to total Lck signal (**A**) or β-actin signal (**B**). Values of the Lck^Y192E^ mice were further normalized to those of Lck^wt^ mice, which were set to 1 for each time point. (**C**) Representative histogram overlays showing the expression of CD3 and TCRβ on thymocytes from Lck^wt^ and Lck^Y192E^ mice (*n* = 4). Graphs on the right side depict statistical analyses of the expression of CD3 and TCRβ. Mean fluorescence intensity (MFI) of CD3 and TCRβ expression from Lck^Y192E^ thymocytes were normalized to MFI values of wild-type cells, which were set to 1. (**D**) Ca^2+^ influx was assessed by flow cytometry upon CD3 stimulation of thymocytes from Lck^wt^ and Lck^Y192E^ mice. Ionomycin was used to show equal loading with Indo-1. One representative experiment is shown (*n* = 6). (**E**) Thymocytes from Lck^wt^ and Lck^Y192E^ mice were stimulated with CD3 antibodies for the indicated time points. Subsequently, cell lysates were prepared, and the activation of Erk1/2 was assessed by immunoblotting using a phospho-specific Erk1/2 antibody. Total Erk1/2 was used to show equal loading. One representative immunoblot from four experiments is shown. The graph on the right side shows quantification of the pErk1/2 signal calculated as described above (**A**,**B**). (**F**) Bar graph represents summary statistics of the activation of Erk1/2 in transitional stage thymocytes determined by intracellular flow cytometry (*n* = 3–5). In all graphs depicted in this figure, each dot represents one mouse, whereas columns show mean values of all analyzed mice + SEM. Statistical analyses were performed using an unpaired Student’s *t* test (* *p* ≤ 0.05, ** *p* ≤ 0.01, *** *p* ≤ 0.001, **** *p* ≤ 0.0001; where not indicated, the values were not statistically significant).

## Data Availability

The data presented in this study are available on request from the corresponding authors.

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
