# Peer review of "Y192 within the SH2 Domain of Lck Regulates TCR Signaling Downstream of PLC-γ1 and Thymic Selection"

_ijms, 2022, doi:10.3390/ijms23137271_

Round 1
Reviewer 1 Report
In the present work, Kaestel et al analysed the role of Lck Y192E mutant in the TCR signalling events regulating thymocyte development and selection. By using knock-in mice they demonstrate that the phosphorylation of Lck in Y192 is essential for thymocyte development since LckY192E mice display an alteration of DN thymocytes with an elevated number of DN3 cells and reduced DN4 cells, thus suggesting a block at the level of pre-TCR. The authors also analysed the early and downstream TCR signalling by evidencing that PLC-g1 phosphorylation and activation is not affected in Lck Y192E mice, whereas Ca2+ influx and ERK1/2 activation are impaired. This work revealed a novel function of Lck in regulating thymocyte maturation by highlighting the pivotal role of Y192.
Major points:
Figure 1. (panel B) Please add the statistical test and significance and specify if the three independent experiments correspond to three mice. Moreover, in addition to pLck, a graph of ERK1/2 densitometric analysis should be added.
Figure 7. (panel A, B) A graph of densitometric analysis of pLck (Y505) and pLCK (394) (panel A) as well as of pTyr, pZAP-70, pPLC-g1 and pLAT (panel B) from the four experiments cited in the legend should be added together with statistical analysis and significance. (panel C) A graph of % of TCR and CD3 expression in Lck wt and Lck Y192E from all mice analysed should be shown together with the statistical analysis and significance. (panel E) A graph of densitometric analysis of pERK from four experiments cited in the legend should be added together with statistical analysis and significance
Reviewer 2 Report
The manuscript is well-written and has some considerable interest to the researchers in the field.
It is quite difficult for a reader not familiar with transgenic TCR models to understand the significance of data summarized in Figures 5 and 6. Please consider to add some background information about this models.
Round 2
Reviewer 1 Report
The authors addressed all the confers.